# Learning Coulomb diamonds in large quantum dot arrays

Oswin Krause[1], Anasua Chatterjee[2],
Ferdinand Kuemmeth[2] and Evert van Nieuwenburg[2]

**1** Department of Computer Science, University of Copenhagen,
Universitetsparken 5, 2100 Copenhagen, Denmark
**2** Center for Quantum Devices, University of Copenhagen,
Universitetsparken 5, 2100 Copenhagen, Denmark

## Abstract

We introduce an algorithm that is able to find the facets of Coulomb diamonds in quantum dot arrays. We simulate these arrays using the constant-interaction model, and rely only on one-dimensional raster scans (rays) to learn a model of the device using regularized maximum likelihood estimation. This allows us to determine, for a given charge state of the device, which transitions exist and what the compensated gate voltages for these are. For smaller devices the simulator can also be used to compute the exact boundaries of the Coulomb diamonds, which we use to assess that our algorithm correctly finds the vast majority of transitions with high precision.


## 1   Introduction

Semiconductor qubits controlled by gate electrodes provide a promising route to large scale quantum computation [1]. In these types of systems, individual electrons are confined in quantum dots (QDs) and qubits are formed, for example, by the spin or charge degrees of freedom of the electrons. Controlling and manipulating the quantum state of the qubits is then achieved by applying external voltages via gate electrodes (gate voltages). By carefully tuning these gate voltages, transitions between different states can be realized, which can then serve as the computational states for quantum computations. The fabrication of large arrays (currently going up to 16 quantum dots [2–5]) creates a new challenge, as the simultaneous manual tuning of many gate voltages quickly becomes infeasible for such large arrays.

Changing the gate voltages and measuring the quantum dot occupations in the systems ground state leads to a so-called charge-stability diagram (CSD), mapping the high-dimensional voltage space to that of the number of electrons on each dot. Constructing a CSD is typically done by performing many two-dimensional raster scans of pairs of gate voltages [3,6]. Based on these raster scans higher-precision line-scans are performed around areas where the QD occupations change to estimate the normal of the charge transition. Knowing the location and normal of the transitions allows one to define compensated gate voltages (i.e. linear combinations of voltages, see below). While this approach is feasible for small arrays of quantum dots, mapping out the CSD for large arrays quickly becomes a very tedious task. To date, hand-tuned loading and shuttling of electrons in QD arrays has been achieved with as large as 8 or 9 quantum dots [2–4]. Automating this process instead is highly desirable, freeing up valuable time towards more impactful experiments. Moreover, since the number of control voltages grows linearly with the number of quantum dots, hand-tuning their values becomes more and more challenging due to cross-talk between the dots. Only recently did we see the emergence of automatic tuning algorithms, often implemented using machine-learning [6–12]. These approaches were used only for small arrays, and still lag behind the results achievable via manual tuning.

In this article, we introduce an algorithm for the automated exploration of large quantum dot arrays, focusing on the identification of transitions between different charge states without the need to label transitions. In particular, the algorithm is designed to reliably find single- and multi-electron charge-state transitions in large quantum dot arrays using simple measurement primitives. The starting point for our tuning problem is a device that has undergone initial tuning: barrier gate voltages are chosen such that individual dots have been formed and sensor dots are tuned to compensate for cross-talk from the individual gate voltages. Moreover, the

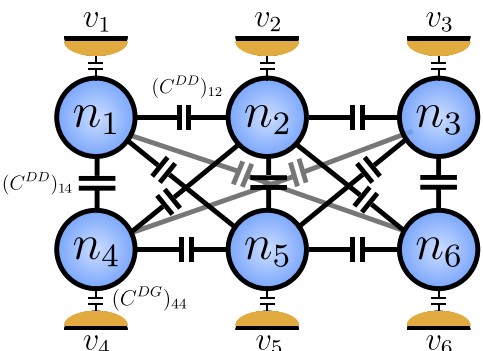

Figure 1: Schematic of a $2 \times 3$ array of capacitively coupled quantum dots, each occupied by $n_i$ electrons, capacitively coupled to external gate voltages $v_j$ described by a matrix $C_{ij}^{DG}$, and coupled to each other via capacitances $C_{ik}^{DD}$. For simplicity of visualisation, the couplings $C_{ij}^{DG}$ are only shown for $i = j$. In this device, the number of gate-voltages $G$ equals the number of dots $N = 6$.

state-space of the device is explored such that initial states of interest are found (e.g., one electron on each dot). With this, we only rely on the use of line scans to detect the nearest transition in a chosen direction, and use a machine-learning model to estimate transitions from measurements. Compared to previous works on the same tuning step [8,9], we will also make use of explicit knowledge of the underlying physics to constrain the problem to make the algorithm more efficient. We restrict ourselves to quantum dot devices described by the constant interaction model [13], appropriate for instance for arrays of gate-controlled spin qubits. In such quantum dot arrays, charge configurations are dictated by Coulomb energies (capacitive coupling), and not by relatively weak tunnel coupling. We further assume that in such a device, we can realize and measure transitions between charge-states reliably up to some precision $\delta$.

The algorithm presented in this work is theoretical. Our goal is to answer the question whether the tuning problem of identifying desired single-and multi-electron transitions is reliably possible for large-devices that follow the constant-interaction model exactly. This is a different starting point than the work in [9] that aimed to develop a practical algorithm that works on small devices that do not require exact adherence to the constant interaction model, but ultimately does not scale to the devices considered in this work.

In the rest of this article we first introduce quantum dot arrays more formally, and proceed with describing the algorithmic solution. We then analyze several relevant scenarios of device layouts and demonstrate the algorithm's performance. The outlook then considers possible relaxations of the assumptions we make along the way.

## 2 Quantum dot arrays and Coulomb diamonds

We consider devices of $N$ quantum dots with significant charging energies ($E_c \gg k_B T$) that are weakly-tunnel coupled to each other and to $G$ external gate electrodes, such as the one schematically shown in Fig. 1. The gate voltages define a high-dimensional space that can be divided into regions where the device assumes a certain ground state of QD occupations. These extended regions are referred to as Coulomb diamonds, because it is the Coulomb blockade effect that prevents additional electrons from tunneling onto a quantum dot unless the potential is large enough [14]. The Coulomb diamonds and transitions between them can be described well using the constant interaction model [13], which, given gate voltages

$v = (v_1, \ldots, v_G) \in \mathbb{R}^G$ and quantum dot occupations $n = (n_1, \ldots, n_N) \in \mathbb{N}^N$, assigns the system a free energy given by:

$$F(n,v) = \frac{1}{2}(|e|n - C^{DG}v)^T (C^{DD})^{-1}(|e|n - C^{DG}v). \tag{1}$$

Here, $|e|$ is the charge of an electron, $C^{DG} \in \mathbb{R}^{N \times D}$ is a matrix whose entries $C_{ij}^{DG}$ store the capacitance between the $i$th dot and $j$th gate, and $C^{DD}$ is a matrix whose off-diagonal elements are inter-dot capacitances. The diagonal elements of $C^{DD}$ are chosen such that $\sum_j^N C_{ij}^{DD} - \sum_k^G C_{ik}^{DG} = 0$.

A Coulomb diamond is then a volume $P_n$ in the high-dimensional voltage space for which the system ground state assumes a set of occupations $n$:

$$
\begin{aligned}
P_n &= \{v \mid \arg\min_r F(r,v) = n\}. \\
&= \{v \mid F(n,v) - F(r,v) \le 0, \; \forall r \in \mathbb{N}^N\}. \\
&= \left\{ v \mid (r-n)^T \underbrace{(C^{DD})^{-1} C^{DG}}_{A} v + b_{nr} \le 0, \; \forall r \in \mathbb{N}^N \right\} \\
&\approx \left\{ v \mid t^T A v + b_{n,n+t} \le 0, \; \forall t \in \{-1,0,1\}^N \right\}. \tag{2}
\end{aligned}
$$

These inequalities describe high-dimensional half-spaces, whose boundaries form the facets of a convex polytope $P_n$. That is, for a given polytope $P$ with equations $\{W_k v + b_k \le 0, k = 1, \ldots\}$, the facets are defined by the sets $\{v \in P \mid W_k v + b_k = 0\}$, $k = 1, \ldots$. The $W_k$ define the normals of the facets, and the $b_k$ their offsets from the origin. In Eq. 2, the term $b_{nr} = \frac{|e|}{2}(n^T (C^{DD})^{-1} n - r^T (C^{DD})^{-1} r)$ does not depend on the gate voltages $v$. The vector $t = r - n$ describes the *changes* in occupation of the individual quantum dots when crossing a facet, and so the inequalities can be intuitively understood as representing a transition $t$ from a state $n$ to a state $n + t$. In the rest of this manuscript we limit the elements $t_i \in \{-1,0,1\}$, because multiple electrons entering or leaving a dot simultaneously is a process that can be neglected in practise.

Of special importance is the polytope $P_0$, the set of boundaries of the state with zero electrons on all dots. In this polytope, all transitions add a single electron on a quantum-dot, i.e. the $t$ are unit vectors, and the normals of the polytope facets are the (scaled) rows of the matrix $A$. Knowing the (scaled) matrix $A$ allows one to compute compensated control voltages $u$, which are related to the gate voltages via a transformation matrix $v = Uu$. The coordinate system of $u$ is chosen such, that $u_i$ only affects the potential of dot $i$. Knowledge of $u$ allows the device user to mitigate capacitive cross talk within the array, which simplifies the task of tuning the device. Geometrically, when measured in coordinates of $u$, the normal of the transition that adds one electron to the $i$th dot is parallel to the $i$th standard basis vector.

In the constant interaction model, the compensated control voltages can be computed by choosing $U$ such, that $A \cdot U$ is a diagonal matrix, for example by choosing $U$ as the (pseudo-)inverse of $A$. This can be verified by inserting the definition of $u$ into equation (2), leading to

$$P_n = \left\{ v \mid t^T A U u + b_{n,n+t} \le 0, \; \forall t \in \{-1,0,1\}^N \right\}.$$

When $t$ is the $i$th standard basis vector, then $tAU$ is just a multiple of $t$. The compensated control voltages are sometimes referred to as virtual gates in the literature [15] and use of this technique was paramount in the recently achieved hand-tuned control over 8 and 9 qubits [2, 3].

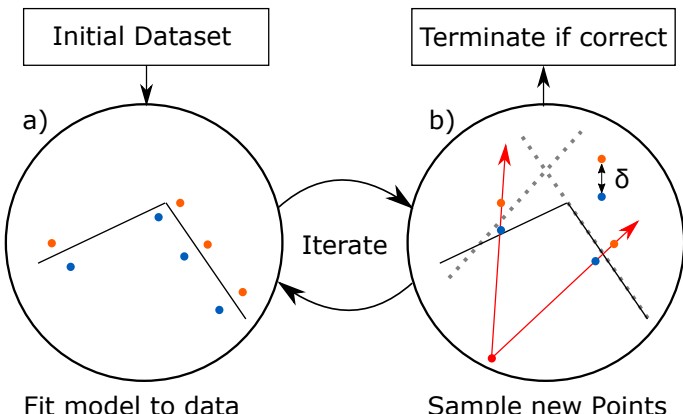

Figure 2: Graphical depiction of the algorithm for a double dot system. Starting with an initial dataset, the algorithm repeats steps a) and b) until termination. a) Algorithm fits model (black lines) to separate points inside the target state (blue) and outside (orange). b) Line-searches are performed from a common starting point (red dot) in the direction of a point of each facet of the model (red arrows). Line-searches produces new pairs of samples (orange/blue dots) with the true boundary (dotted lines) in-between. The distance between red and blue dots is smaller than $\delta$. See text for more details.

## 3   Method

For our method, we assume that the device follows the constant interaction model and that it is equipped with non-invasive charge sensors that can detect the occurrence of a transition reliably without influencing the underlying capacitances. Further, we assume that the user can provide a line-search procedure that performs a raster scan along the line between two points of gate-voltages $v_{\text{start}}$ and $v_{\text{end}}$, and uses the charge sensors to detect the existence of a transition reliably along this line. We further assume, that the user can find two points $v^-, v^+$ along this raster scan that bound the position of the transition (i.e. $v^-$ and $v^+$ lie on opposite sides of a transition) and that the user can give a bound of the maximum distance $\|v^- - v^+\| \le \delta$.

Our task is to estimate the facets of a polytope $P_n$ as best as we can, so that we can identify the voltage regions corresponding to fixed quantum dot occupations. While $P_n$ has, in the worst case, $3^N - 1$ facets, in practice only a small set of transitions is of interest. Our algorithm therefore starts with a pre-selected list of such transitions that are of relevance to an experiment. For example, we may ask for all single-electron transitions from a specific charge state (e.g. $P_0$), or transitions that leave the number of electrons unchanged. In many applications, the set of all "one-electron transitions" is relevant, i.e., transitions in which exactly one electron moves from one dot to another, or from a reservoir to a dot (or vice versa).

We do not assume that the list is an exhaustive enumeration of all transitions present in the target polytope, nor that all transitions on the list exist. The algorithm will then use data gathered by the line-searches to select a subset of existing transitions from the list and produce a new set of candidate line-search directions in order to refine the dataset and fit the polytope again. This process is repeated until all transitions selected are well-supported by the measurements and the transitions which are not selected are similarly ruled out with high certainty.

The algorithm can be described by alternating two steps, depicted in Figure 2. First, an initial dataset $\mathcal{D}$ of measurements is provided by the user. The dataset consists of pairs of points $(v_i^-, v_i^+)$, $i = 1, \dots, \ell$, bounding a transition of interest. These pairs of points are acquired by

the aforementioned line-search procedure. The initial dataset can contain data from line-searches acquired by picking random directions, but may also include educated guesses of informative search directions.

The algorithm then proceeds to estimate the set of transitions using a regularized maximum-likelihood approach (see section 3.1 below). This estimation can be erroneous and may represent a local optimum, or the initial dataset may have a flaw that makes estimation difficult, e.g., it does not feature enough data points on each transition to estimate it reliably. Because of this, we use the estimated model to acquire additional points. We sample a point on each estimated facet and then perform a line-search from a point inside $P_n$ through this point (see Figure 2b and Appendix A).

If the estimated model is correct, the points returned by the line-search will bound the estimated transition. Otherwise, they are an example of an error of the model and can be used to update the estimate. In either case, they are added to $\mathcal{D}$ and a new model is estimated. This process becomes slightly more complicated due to the existence of transitions for which the model has low confidence or which are not supported by points in the current dataset. For these transitions we analyze the learned model and create new candidate facets which we use to produce informative measurement directions that proof or disprove the existence of these facets (see Appendix A).

This loop is repeated until we are confident that all transitions found are either correct or too small to be estimated reliably. For the former, we count the number of pairs $(v_i^-, v_i^+)$ that are separated by a transition and for the latter, we use the radius of the largest $G-1$ dimensional hypersphere that can be inscribed into the facet. For details, we refer to Appendix B.

In the following, we describe the model we have chosen to represent and learn a polytope and how we make it practical for searching relevant transitions of a given state. We will focus on the general idea of the model, and provide a running example of the algorithm, while the detailed descriptions of the steps are given in the appendices. Further, a reference implementation of the algorithm is available at [16].

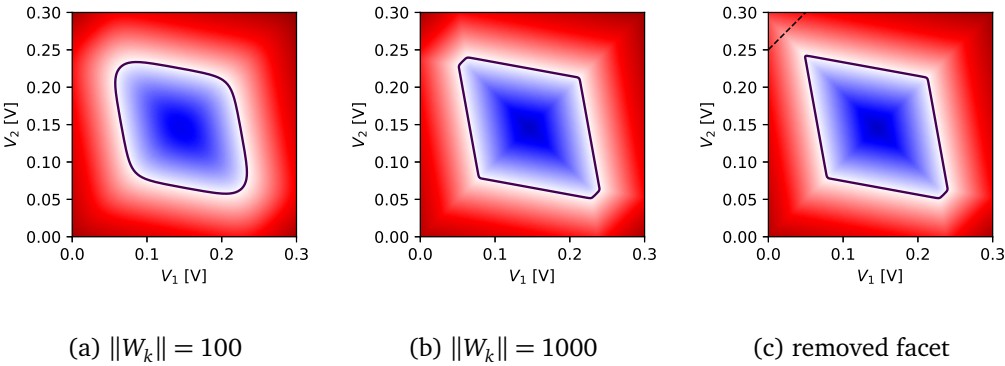

(a) $\|W_k\| = 100$       (b) $\|W_k\| = 1000$       (c) removed facet

Figure 3: Visualization of classifier $h(v)$ for several choices of model parameters. Black lines indicate the contour $h(v) = 0$ while blue (red) pixels indicate $h(v) < 0$ ($h(v) > 0$). Figures a) and b) use scaled parameters of $W_k$ and $b_k$ from a polytope of the $(1,1)$ state of a simulated double dot. Parameters are scaled such that the norms $\|W_k\|$ have fixed values. For small norms in a), $h(v) = 0$ has a rounded shape, while for larger norms in b) the shape $h(v) = 0$ approximates the underlying convex polytope well. Figure c) demonstrates the removal of a facet by moving it outside the polytope (dashed line).

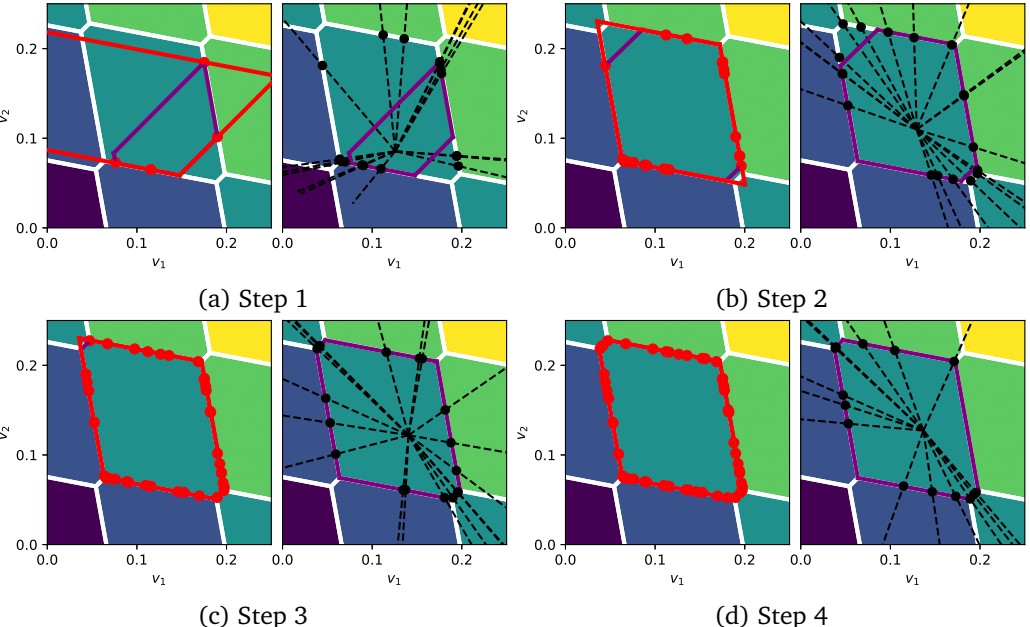

(a) Step 1          (b) Step 2

(c) Step 3          (d) Step 4

Figure 4: Visualization of a toy example of the algorithm for learning the polytope of the (1,1) state on a simulated double-dot device. Colored background with white lines represent the different states and boundaries of the device. Background colors are chosen based on the total number of electrons occupying the device in the state. a-d) Left: Model (red lines) fit to the measurements (red points, only $v^-$ are shown for ease of visualization). Purple lines: adapted model for sampling to account for facets that were not found during the fit. Right: Sampling process. for each facet of the adapted model three line-searches (dashed lines) are performed creating new pairs of datapoints (black points).

## 3.1 Estimating the model

To learn a polytope $P_n$ with equations $W_k v + b_k \leq 0$, $k = 1, \ldots, M$ we have to identify the correct set of transitions and then estimate their parameters. We start with a dataset of point pairs of gate voltages, $\mathcal{D} = \{v_i^+, v_i^- \in \mathbb{R}^G, i = 1, \ldots, \ell\}$ with $\|v_i^+ - v_i^-\| \leq \delta$, and assume that $v_i^-$ belongs to the state $n$, while $v_i^+$ belongs to a different state.

We formalize learning the polytope from data as a binary classification task, where the goal is to find a classifier $h$ that separates all $v_i^-$ from $v_i^+$. We derive this model classifier from a multi-class logistic regression model, the construction of which is fully contained in Appendix D:

$$h(v) = \log \sum_{k=1}^{M} \exp(W_k^T v + b_k). \tag{3}$$

A point is deemed as inside the polytope if $h(v) < 0$, and outside otherwise. This model is well suited for modeling convex polytopes as, when $\|W_k\|$ becomes large, the set $\{v \in \mathbb{R}^G | h(v) \leq 0\}$ approximates a convex polytope arbitrarily well. This is demonstrated in Figures 3a&3b. Figure 3c demonstrates that an unnecessary facet can be removed from the model by choosing its offset $b_k$ such, that it is moved away from the polytope.

It is important to note at this point that, while we reuse the name $W_k$, there are subtle differences in their interpretation compared to the earlier use in the polytope $P$. For the polytope $P$, the norm $\|W_k\|$ can be chosen arbitrary, while for the probabilistic model the norm carries additional information: the larger the norm, the more the transition is supported by the data,

and the sharper the probabilistic model becomes. This also means, that after model fitting, transitions with small norms need to be handled separately.

While simple, this initial adaptation of the polytope $P$ as statistial model is not practical: The number of parameters of $W$ is large and there is no easy way to infer which transition is modeled by a given row $W_k$. We can considerably improve on this by making use of physical knowledge of the transitions owing to the constant interaction model as follows. In the constant interaction model, following (2), each transition has a normal $W_k \propto t_k A$. Thus, if $A$ were known, $W$ could be computed easily.

It is possible to estimate a row $A_k$ up to a factor by identifying a transition that adds an electron on the $k$th dot and measuring its position and normal (in voltage space). However, the length of the normal vector remains unknown. To represent this, we decompose $A = \Lambda \Gamma$, where $\Lambda$ is a real diagonal matrix and $\Gamma$ stores the estimated normal vectors. This decomposition is not strictly necessary from a mathematical point of view, since every change of the norm can be folded into Gamma by multiplying its rows with a diagonal entry of Lambda. However, during optimization, it allows us to differentiate learning the direction of the transitions ($\Gamma$) and their norm ($\Lambda$). In our approach, both aspects are handled independently, as we have different prior knowledge regarding the distribution of directions and their relative norm differences. We will describe this in more detail in the following.

The matrix $\Gamma$ can be learned efficiently, by measuring the transitions of the polytope $P_0$. This polytope has exactly $N$ transitions, one for each dot, and the transitions are $t_k = (0, \ldots, 0, 1, 0, \ldots)$, i.e. the unit vector with a 1 at index $k$. Consequently, $t_k^T A = A_k$, and hence identifying the transitions of $P_0$ gives a direct estimate of $\Gamma$. The entries of $\Lambda$ can then be estimated by measuring transitions of electrons from one position in the array to the next. Moving an electron from position $i$ to position $j$ leads to a transition with normal $t^T A = A_j - A_i$. Using an estimate of the normal, we can find $\Lambda$ such, that $t^T A \propto t^T \Lambda \Gamma = \Lambda_{jj} \Gamma_j - \Lambda_{ii} \Gamma_i$.

This logic needs to be translated into our fitting code and model Eq. (3). To obtain an estimate of $\Gamma$, we set $W = \Gamma$, and train the model using regularized maximum likelihood using data $v_i^-, v_i^+$ obtained from the device initialized in state $n = 0$. This problem is non-convex and highly multi-modal, often resulting in getting trapped in bad fits, especially with only a small number of data points. For the minimization therefore, we include a regularizer $\Omega(\Gamma)$ that steers the optimizer away from the bad local optima using prior knowledge of the task. Knowledge of the physics of the system comes in at this step, where we make use of two properties that are present in common device designs:

1. For all transitions, we expect the absolute voltage value required to add an electron on the $k$th dot to be small

2. We assume that the capacitive coupling between a dot (i.e. its plunger gate) and its gate electrode dominates the influence of all other gate electrodes. That is, the angle between the normal of the transition adding an electron on the $i$th dot and the coordinate axis of the corresponding gate voltage is small.

More details on how these are implemented are in Appendix E.

Having estimated $\Gamma$, we can now proceed with the estimation of $W$ (and $b$) of the target polytope $P_n$. We decompose $W = cTA = cT\Lambda\Gamma$, with $T$ a matrix whose rows are the transitions $t_k$ we are interested in, and $c$ a real diagonal matrix that enables sharpening the polytope (see Figure 3b). A single transition is thus described as

$$W_k = c_{kk} \sum_{i=1}^{N} t_{ki} \Lambda_{ii} \Gamma_i. \tag{4}$$

Together with a good estimate of $\Gamma$, we can now learn $P_n$ using regularized maximum likelihood given the pairs of data $v_i^-, v_i^+$ obtained from line-searches with the device initialized

in state $n$. For simplicity, we assume that the rows $\Gamma_k$ are normalized s.t. $\|\Gamma_k\| = 1$. With this, we keep $\Gamma$ fixed and only find the values of the diagonal entries of $c$, $\Lambda$ and the vector of offsets $b$. We again use regularized maximum likelihood on the probability of assigning the right class label to the points in a point-pair. Here, too, we add a regularizer that we define in more detail in Appendix E.

Due to the multi-modality of the solution, the optimisation performance varies a lot with the initialization for parameters $c, \Lambda$ and $b$. If we have no prior knowledge of a possible shape of the polytope (e.g., in the first iteration), we pick $c_{kk} = 1/\delta$, $\Lambda = I_N$ and $b_k = -c_{kk} \max_i^\ell W_k^T v_i^-$. This choice of $b_k$ ensures that all points $v^-$ are classified correctly by the initial model and each facet is close to points in the dataset. If we already have a previous estimate, we choose $\Lambda$ as the same value from the previous iteration.

## 3.2 Example

We are now ready to describe one example run of the algorithm. In this example, our goal is to find the transitions of the $n = (1,1)$ state in a double-dot device. We assume that the algorithm has already been used to learn $\Gamma$ from $P_0$. In the $(1,1)$ state, six transitions are possible: four for adding or removing an electron on a dot ($t_k = \pm(1,0)^T$, $t_k = \pm(0,1)^T$), and two for moving an electron from one dot to another ($t_k = \pm(1,-1)^T$). Thus, $T$ can be chosen as a 6x2 matrix with these transitions as rows. After choosing an initial set of four points, the algorithm proceeds to produce the steps given in Figure 4.

In the first step, the algorithm fits the model (4) to the initial dataset (Figure 4a, Left, red lines). Since the initial set of points is small, many transitions could not be found. To sample informative points for these transitions, the estimated values of $\Gamma$ and $\Lambda$ from the model-fit are used to add the missing facets at positions such that all $v^-$ are correctly positioned inside the polytope (See Appendix A). The resulting polytope (Figure 4a, purple lines) is used to generate line-search directions (dashed lines) by sampling three points on each facet. The line-searches result in new measurements of the boundaries (black dots). These points are added to the dataset and then another model is fit (Figure 4b, Left, red lines). Now the model correctly identified four facets and only the two small transitions are missing, which are then added again for sampling (purple lines). This is repeated several times until in Step 4, Figure 4d all facets have been identified. From this point onward, the algorithm continues sampling line-search directions on all facets which are not supported by enough datapoints, until there are 5 or more points on each facet, at which point the algorithm terminates.

## 4 Results

We consider six different scenarios to test and demonstrate our algorithm in. Each scenario is characterized by a different 2D device geometry (c.f. Figure 1) and a different set of target states or transitions. In all examples we assume for simplicity that the number of gates equals the number of dots and thus $G = N$. For each scenario we generate devices with different capacitance matrices $C^{DD}$ and $C^{DG}$, using the scheme given in Appendix F. The scheme depends on a parameter $\rho$ that changes the interaction strength (or cross-talk) between gates and dots. We generate 20 devices with weak ($\rho = 1$) and strong ($\rho = 3$) interaction strength, and estimate their ground truth polytopes (see Appendix G). The interaction strengths are chosen such that the polytopes obtained from $\rho = 1$ devices are similar to many practical devices, while for $\rho = 3$, the resulting polytopes have large distortions, which are likely larger than encountered in practice and more difficult to estimate, especially for the task of estimating $\Gamma$. The generated device parameters are chosen such that the gate-voltages are measured in

V and the size of the polytopes are around 0.2 (i.e., a range of 200mV) along each axis. We consider the following scenarios:

$S_1$ As an example for a sequence of operations involving several states, we select a 3x2 device and aim to find all transitions needed to exchange a pair of electrons. This exchange is done in such a way that spin can be preserved and thus electrons can not be positioned on the same dot, due to the Pauli exclusion principle (i.e., we assume that the electrons are not known to have different spin). We choose a sequence of transitions that requires computing polytopes for the following 6 states (the corresponding transitions $t$ are just the differences between consecutive states). Colors indicate the intended electron positions after an operation:

$$
\begin{pmatrix} 1 & 0 \\ 0 & 0 \\ 1 & 0 \end{pmatrix} \rightarrow
\begin{pmatrix} 1 & 0 \\ 1 & 0 \\ 0 & 0 \end{pmatrix} \rightarrow
\begin{pmatrix} 1 & 0 \\ 0 & 1 \\ 0 & 0 \end{pmatrix} \rightarrow
\begin{pmatrix} 0 & 0 \\ 1 & 1 \\ 0 & 0 \end{pmatrix} \rightarrow
\begin{pmatrix} 0 & 0 \\ 0 & 1 \\ 1 & 0 \end{pmatrix} \rightarrow
\begin{pmatrix} 0 & 0 \\ 1 & 0 \\ 1 & 0 \end{pmatrix} \rightarrow
\begin{pmatrix} 1 & 0 \\ 0 & 0 \\ 1 & 0 \end{pmatrix}.
$$

Here, for each computed polytope, $T$ contains all transitions that add or remove an electron as well as all transitions of a single electron to a different location.

$S_2$ As an example for a more complex transition involving multiple electrons, we consider a 3x2 device with 3 electrons in a zig-zag configuration. The goal is to find the transition that lets all electrons simultaneously change side, i.e. we need to find the transition

$$
\begin{pmatrix} 1 & 0 \\ 0 & 1 \\ 1 & 0 \end{pmatrix} \rightarrow
\begin{pmatrix} 0 & 1 \\ 1 & 0 \\ 0 & 1 \end{pmatrix}, \; t =
\begin{pmatrix} -1 & 1 \\ 1 & -1 \\ -1 & 1 \end{pmatrix}.
$$

To find this facet reliably, we need to include most facets that are near it or intersect with it. However, we don't want to include all transitions, since on larger devices, the total set of all possible transitions would be too large. As a result, we choose $T$ to include the following transitions:

- $t$ and transitions that can be obtained from $t$ by setting one entry to 0

- Transitions that affect up to two electrons (and 4 locations in the array) simultaneously.

The second group includes transitions in which up to two electrons move from one dot to another, or from a reservoir to a dot (or vice versa).

$S_3$ In a 3x3 device with one electron on each dot, find all transitions that add, remove or move a single electron. $T$ is chosen using the same principle as in $S_1$.

$S_4$ The same scenario as $S_3$ with a 4x4 device.

$S_5$ The same scenario as $S_4$ but we restrict $T$ further to only include pairwise transitions between direct neighbours. This means we exclude for example searching for a transition of an electron from the top-left to the bottom right corner. This reduces the number of transitions considered from 284 in $S_4$ to 112.

$S_6$ We consider a 4x4 device with one electron on each dot. However, in this case we assume that we can completely detach the reservoir and thus the device can not exchange electrons with it. Technically, this means that the computed polytopes no longer represent ground states of the device.

This way only transitions are possible that keep the amount of electrons constant, i.e., transitions $t$ that move electrons within the array, thus $t^T \mathbb{1}_N = 0$ (here, $\mathbb{1}_N$ denotes the all-one vector with $N$ entries). These polytopes are special as they are unbounded independent of the state the device is in, which we show in the following. Assuming $t^T \mathbb{1}_N = 0$, we can compute the normal of a transition $W_k = t_k^T A$. If we take the direction $q = A^{-1} \mathbb{1}_N$, we obtain

$$W_k^T q = t_k^T A A^{-1} \mathbb{1}_N = t_k^T \mathbb{1}_N = 0 \,.$$

Thus, the normals of all facets of the polytope are orthogonal to the direction $q$. From this follows, that for any point $v \in P_n$, $v + tq \in P_n$, $\forall t \in \mathbb{R}$.

To solve this issue and to obtain bounded polytopes, after estimating $\Gamma$, we choose $N-1$ independent vectors orthogonal to $\hat{q} = \Gamma^{-1} \mathbb{1}$ and find the projected polytope in the resulting $N-1$ dimensional subspace. This projection can be computed using a householder reflection matrix that maps the direction $q$ to the first basis vector that can then be discarded. While the shape of the resulting polytope depends on the chosen $\Gamma$, the polytope will always represent a finite polytope, as long as $q^T \hat{q} \neq 0$.

For $T$, we consider all single electron inter-dot transitions in this example. Due to the fact that the large transitions that include exchanges with the reservoir cannot take place in this scenario, all $N \cdot (N-1) = 240$ such transitions are present on the ground truth polytope.

In all scenarios, we use our algorithm to estimate $\Gamma$ and the relevant polytopes, and use two different choices, $\delta \in \{0.001, 0.002\}$, for the line-search precision. To prevent endless runs we terminate either when, for computing $\Gamma$, more than 4000 line-searches were conducted, or if for the individual polytopes in a scenario more than 15000 line-searches were conducted. As lower bound for voltages considered by line-searches, we take $v_i \geq -2$. As a lower bound for the size of a facet to be considered we require the radius $r$ of the largest hypersphere that can be inscribed into it to be $r \geq r_{\min} = 2\delta$.

As initial datasets, for estimating $\Gamma$ in a device with $N$ dots, we sample $4N(N+5)$ initial points, a number which worked well empirically in our experiments. As starting point, we choose $v = -2\mathbb{1}_N$, the lower bound in all directions and we sample in direction $p = \exp(2y)$, where $y \in \mathbb{R}^G$ is a standard normal variable. This prevents sampling in directions that can violate the lower bound. This strategy is designed to give enough informative measurements that allow the algorithm to terminate quickly.

For the target polytopes $P_n$, we sample $N^2$ initial points. We first create a starting point for line-searches by sampling a point close to the boundary. This is uninformative and thus reflects the fact that we often have no good estimate for the center of the polytope. Then we sample direction vectors uniformly on the unit sphere. This setup only gives little information of the small facets of the polytope, and thus finding the small facets purely relies on our ability to sample informative candidate points during the run of the algorithm.

Next, we present our evaluation of the results. We will focus here on measurements for precision and accuracy of the estimated polytopes. Our results on running time of the algorithm and required number of line-searches can be found in Appendix H.

## 4.1 Evaluation of $\Gamma$

We first evaluate the results of our algorithm for computing $\Gamma$. For this, we evaluate the values of $\Gamma$ computed in scenarios $S_1$ (3x2), $S_3$ (3x3) and $S_4$ (4x4). We first evaluated in how many cases the algorithm succeeded. This means that the algorithm terminated and for each row of $\Gamma$, we found at least $N+3$ pairs of samples separated by it.

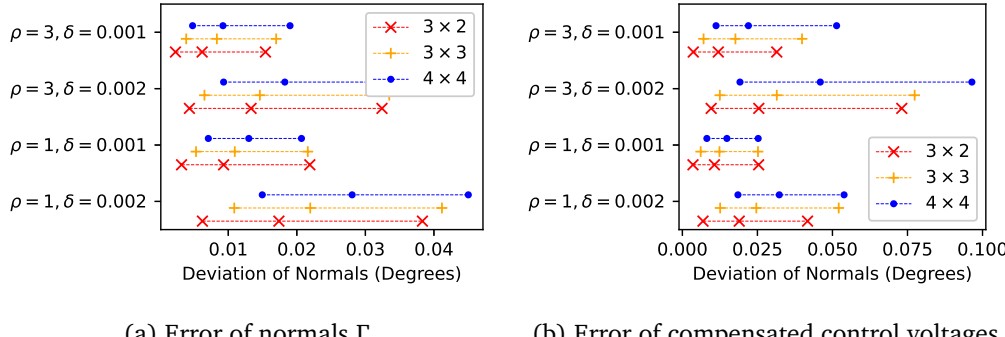

(a) Error of normals Γ

(b) Error of compensated control voltages

Figure 5: Accuracy of the algorithm for estimating normal vectors Γ. We show both the angular deviations of the estimated rows of Γ to the true values as well as the misalignment of the compensated coordinate system.

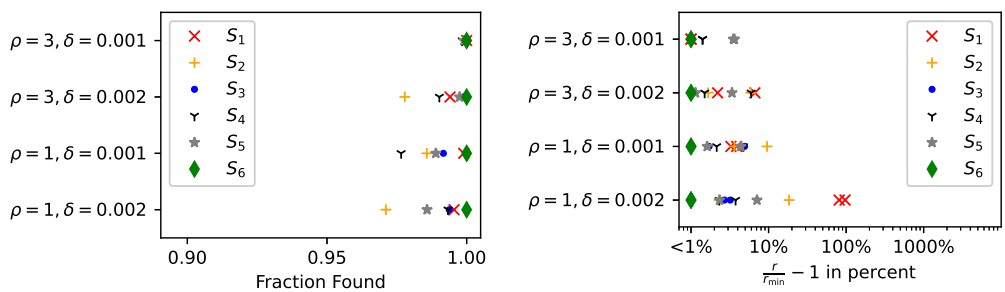

(a) Fraction of target transitions found

(b) Relative size of missed transitions

Figure 6: Success rate of the algorithm for computing $P_n$. Left: fraction of facets found. We take only facets of the polytope into account that are included in $T$ (e.g., "all one-electron transitions") and would be selected by the stopping criterion. Right: for missed facets, we measure the relative size compared to our stopping criterion. See the text for more details.

In all our experiments, we only observed failed trials on the 4x4 device with setting $\rho = 3$, where the solver could get trapped in a local optimum. This happened in approximately 10% of trials. This also means that for these trials no other polytopes of the scenario could be computed and we exclude them from our experiments. Still, all our results reported for $S_4$, $S_5$ and $S_6$ had at least 17 successful trials for $\rho = 3$ (out of 20 trials).

For the successful trials, we computed the angular differences between $A_k$ and $\Gamma_k$ for all dots and trials and report their 5%, 50% and 95% quantiles of the angular errors in all scenarios in Figure 5a. In all cases, the error is below 0.1 degrees. Further, we measured the correctness by computing the angle between the transitions of $P_0$ in the coordinate system of the compensated control voltages (i.e. the angle between the rows of $A\Gamma^{-1}$) and the standard basis. This is the most important metric, since it measures how perpendicular the transitions of $P_0$ are in this coordinate system and thus, how well the compensation works. Again, we report their 5%, 50% and 95% quantiles in all scenarios in Figure 5b. The error is slightly higher but still below 0.15 degrees.

## 4.2 Evaluation of $P_n$

Next, we evaluate the estimated polytope transitions in the six scenarios. Our evaluation focuses on three different error conditions:

**False Positive** The algorithm returns a transition that does not exist on the device.

**False Negative** The algorithm misses to learn a facet that does exist on the device, is included in $T$ and has largest inscribed hypersphere radius $r \geq r_{\min}$.

**Unusable transition** An existing transition is found, but the estimates are so poor that using it results in a different transition.

For false negatives, the condition on size might sound odd at first. After all, we would hope to find all transitions in $T$ that exist in the polytope. However, our line-searches operate with limited accuracy, making it impossible to detect facets that are smaller than the accuracy of the line-search.

This is reflected in the design of the algorithm, as it only returns facets with largest inscribed hypersphere radius $\geq r_{\min} = 2\delta$ (See Appendix B). Consequently, it only makes sense to use the same threshold on the facets of the ground truth polytope to check whether they should be even considered for comparison. As a result, changing $\delta$ effectively changes the task, as not only the precision of the estimated polytope is changed, but also the ground truth we compare it to. For example in $S_4$, with $\delta = 0.002$ we only expect to find approximately 80 facets on average, while at $\delta = 0.001$, this number rises to 100.

To evaluate whether a facet is usable, we take the center of the computed largest inscribed hypersphere on the facets of the estimates polytope and perform a line-search through this point from inside the polytope. We then evaluate whether the facet hit in the ground truth polytope has the same label as the label assigned to the transition by the model. This mimics how a practitioner would use these facets in order to change the ground-state of the device.

In all converged trials, we did not find a single instance of a false positive or of an unusable transition. This leaves evaluation of false negatives. The fraction of facets found in all scenarios is shown in Figure 6a. In all scenarios and for all settings, more than 96% of the transitions have been found and for $S_6$ all facets have been found in all trials.

For the missed facets with radius $r \geq r_{\min}$, we computed the relative size-difference $\frac{r - r_{\min}}{r_{\min}}$ of the radius of the inscribed hypersphere to the stopping criterion. The reported 50% and 95% quantiles are shown in Figure 6b. One can see that for the scenarios $S_3$, $S_4$, $S_5$ and $S_6$ the missing facets have a difference in radius consistently smaller than $1.1 r_{\min}$, while for $S_1$ we observed a few facets being missed with radius $r \approx 2 r_{\min}$ in the case of $\rho = 1$ and $\delta = 0.002$. However, this number is based on a total of 7 missed facets over all estimated polytopes.

For scenarios $S_1$ and $S_2$ we were targeting specific (sequences of) transitions. We evaluated whether the tasks were solved by checking whether all required transitions were found. For $S_2$ the task was to find a single transition. This was achieved in all but 2 trials in the setting $\rho = 1, \delta = 0.002$. The remaining 78 trials succeeded. For $S_1$ we had to check whether all 6 transitions were found. For this we made use of the fact that each transition appears as candidate in two polytopes: either as part of the polytope belonging to the initial state, or as transition in the polytope from the target state back to the initial state. If any of them succeeded in finding the transition, we counted the transition as found. With this strategy, we managed to find the sequence of transitions in all cases but one for $\rho = 1, \delta = 0.002$. In this case, we found that one of the required facets in the ground truth polytope was smaller than our cut-off radius and thus the task was not solvable for the algorithm with our chosen stopping criterion.

## 5 Discussion

In this manuscript we introduced an algorithm for determining Coulomb diamonds (polytopes) in large quantum dot arrays. We used regularized maximum likelihood optimization

to estimate a model (Eq. 3), and tested it on several scenarios. In almost all of the cases, the resulting polytopes and transitions were found with high accuracy. Most importantly, our algorithm did neither return any transition that did not exist on the device, nor mislabeled any of the transitions it found. Moreover, we showed that we can construct a point on each facet such, that a linear ramp from a point inside the polytope through it results in the desired ground state transition.

Additionally, our method for proposing new line-search directions is efficient. Our algorithm required less than 15000 measurements even in devices with 16 dots with up to 100 facets of interest. This places our algorithm within a factor 10 of the minimum amount ($G \cdot \|T\|$) of measurements required to verify the learned polytope. This is in contrast to the bounds obtained for the number of line-search directions for uninformed sampling strategies (see Theorem 2 in [17]), which are exponential in the dimension of the number of dots and which assume perfect line-search accuracy. The reason for our sample efficiency compared to the naive bound is our sampling procedure, as visualized in Figure 4. As the size of the estimated facet is used as part of the sampling process, facets that are believed to be big are sampled with low density, while in small facets, high density sampling is performed. Since the shape of the polytopes are dominated by a few large facets, this allows to save many line-searches in the majority of estimates.

To achieve this, we made several assumptions. First, we use the constant interaction model to describe the array of quantum dots. We also assume that the devices can realize, and measure, transitions between ground-state transitions reliably. This means that we can measure whether an electron moved inside the array and that the device is manufactured such, that existing electron transitions occur when the device control parameters are chosen accordingly. For example, an existing transition adding an electron to a quantum dot requires that this dot can easily exchange electrons with a reservoir and does not need to rely on co-tunneling events. However, this is not a limitation of the algorithm, but of the device. if a theoretically existing ground state transition does not occur during our measurements, we can as well treat the transition as non-existing and we have shown that our algorithm handles missing transitions gracefully.

Our assumptions on the measurements of transitions are weaker. The only assumption we make here is that when linearly ramping the gate voltages between $v_{\text{start}}$ and $v_{\text{end}}$, we can reliably measure whether a transition occurred and that we can determine the transition voltages with within a precision $\delta$. We do not require that we know *which* transition occurred nor how the measurement is conducted. We treat $\delta$ as an external parameter: if the sensor signal is noisy and uncertain, our algorithm will miss facets of the Coulomb diamonds that are small since they can not be detected reliably anymore.

While the devices we consider are idealized, our work is an important stepping stone towards fully autonomous tuning of these transitions in spin qubit arrays. If we fail to find an algorithm under the idealized conditions described in this work, we can not expect to find an algorithm that performs well on a real device. On the other hand, if we find an algorithm that works reliably, there is hope that this algorithm can be generalized to less favourable conditions in the future, or that improvements in material design, manufacturing and sensing progressively narrow the gap. In this case, we have shown that our algorithm provides the essential tools to program a device reliably.

Still, our algorithm is not perfect. For estimating $P_0$, which is important for computing compensated control voltages, we have observed that our algorithm does not find the solution in all cases for large devices. We expect that this problem can be solved simply by re-running the algorithm with a new set of measurements. Failures here are expected, as the problem of estimating convex polytopes using a binary classification dataset is known to be a NP-hard machine-learning problem [18]. Our results also showed that our strategy for obtaining ad-

ditional measurements does not help to solve the issue: In most trials the algorithm managed to find the solution with the chosen initial set of randomly sampled points, or not at all. We suspect that this problem can be solved more efficiently by obtaining measurements not only of the position of the transition but also of its normal, as has been utilized by experimentalists in the literature [3]. However, adapting this approach requires a more careful modeling of the sensor signal, which is outside the scope of our work.

A limitation of our work arises due to the charge sensors themselves. In our algorithm, we do not model the effects of charge sensing on the constant interaction model, thus adding the implicit assumption that charge sensors do not interact with it. This is an assumption that may not be valid in practice, because charge sensors are capacitively coupled to the array and therefore change the capacitance matrices. Still, our results carry over to weakly coupled charge sensors, when their effect on the constant interaction model can be approximated by adapting (2) to

$$P_n \approx \left\{ v \in \mathbb{R}^G \mid t^T A v + t^T A^S v^S + b_{n,n+t} \leq 0, \, \forall t \in \{-1, 0, 1\}^N \right\},$$

where $v^S$ is the vector of gate voltages of the sensor gate electrodes and $A^S$ is the effect of the gate electrodes on the dots of the device due to capacitive coupling. In practice, gate-voltages $v^S$ are chosen to compensate for changes of the control voltages $v$ to keep the potential of the sensor dots constant. This compensation is done by choosing $v^S$ as affine linear functions of $v$, i.e., $v^S = U^S v + b^S$. The resulting polytope as a function of $v$ using sensor compensation is described as

$$P_n^{\text{Comp}} \approx \left\{ v \mid t^T \underbrace{(A + A^S U^S)}_{\tilde{A}} v + \underbrace{b_{n,n+t} + t^T A^S b^S}_{\tilde{b}_{n,n+t}} \leq 0, \, \forall t \in \{-1, 0, 1\}^N \right\}.$$

The polytope still abides to the general form of (2), especially the connection of transition $t$ and its normal, which allows to compute the polytope $P_0^{\text{Comp}}$ to obtain an estimate of $\tilde{A}$ that can be used to learn $P_n^{\text{Comp}}$ with the adapted offset $\tilde{b}_{n,n+t}$. Thus, our proposed algorithm can still be used to learn the transitions of the *compensated* gate voltages, assuming that the sensor compensation function is the same for all polytopes of interest and the number of electrons on the sensor dot remains constant. We leave more complex interaction of sensors for future work.

More generally, future work must incorporate more general types of deviations of the device from the constant interaction model, which requires adaptations of our model and fitting process. While the work presented here is able to adapt to some deviations due to the freedom of picking parameters $b$ and $\Lambda$, deviations in the normals of the learned facets are currently not modeled well. This holds especially for deviations that cause the facets of transitions $t$ and $-t$ (e.g., adding and removing an electron at a dot location) to not be parallel. These deviations can be described by generalisations of the constant interaction model, in which the capacitance matrices $C^{DD}$ and $C^{DG}$ may depend on the occupation numbers $n$ of the array. These changes reflect that dots with large $n_i$ are typically somewhat larger and thus have typically larger capacitances, than a dot with small $n_i$.

In this case, using an estimate of $\Gamma$ obtained in the empty device ($P_0$) might not be a good description of the normals of the coulomb diamond of a state with large electron occupation. However, if we still assume that deviations from parallelity in the polytope $P_n$ are small, we can try to adapt $\Gamma$, while learning the polytope. This requires a careful choice of regularisation and adaptation of the stopping criteria to expect larger errors. For larger deviations, additional terms must be incorporated into the model.

The assumption of linear transitions is strong, as it is unlikely that the device has perfectly linear boundaries between ground-states. However, prior work [9] has already shown in practical application that at least some small devices have coulomb diamonds that are sufficiently

linear to fit a convex polytope to them, showing that the strategy has the potential to become practical on some devices.

Compared to our approach, the approach in [9] used much weaker assumptions on the shape of the polytope, which affects computation time. In their results, computing a polytope for a 2x2 array took 4 hours, while our approach takes less than 1h on a 4x4 array (scenario $S_5$). Further, the approach in [9] needs to compute convex hulls and halfspace-intersections in gate-voltage space, which is a problem that grows exponentially in the size of the array and becomes infeasible already in 4x4 arrays. Finally, our approach offers the possibility to only search for a set of transitions of interest, unlike the model-free approach used in [9], that requires finding all facets of sufficient size.

The possible gains of restricting the search to relevant facets are significant, as our results on the 4x4 devices in scenarios $S_4$ and $S_5$ showed. The polytopes computed had up to 3700 facets of which 1200 are large enough to be considered by our size cut-off of $2\delta$. Of these, only 100 were contained in our set of transitions $T$ considered in $S_4$. The comparison of running times of $S_4$ and $S_5$ reveals that reducing $T$ further not only reduces the computation time, but also the number of line-searches. In the best case, we observed that reducing the number of transitions considered by a factor of 2.5, reduces the computation time by factor of 2-3 and the number of line-searches by 30-50%. In our scenarios, this reduction comes at almost zero cost in the usability of the result, because almost all single-electron transitions exist between direct neighbours. Thus, a smart pre-selection of transitions can significantly reduce running time.

Further, our work does not cover the tuning of additional gate electrodes, for example "barrier electrodes" that tune the tunnel barriers between the dots. We currently assume that practitioners can fix those parameters before using our algorithm for estimating $\Gamma$ and other polytopes $P_n$. However, these electrodes are fine-tuned after certain states have been found (e.g., the state $n = (1, 1, \ldots, 1)$ or are changed during device operation, and thus affect the learned polytopes due to capacitive interactions. A full incorporation of these barrier electrodes in our algorithm is therefore a challenge and a desirable next step to allow automatic tuning of quantum dot arrays.

While our strategy might lead to practical algorithms for some devices, many devices of interest do not fulfill the linear assumption and curvilinear approaches are required. Our work does not offer a direct solution for these devices but there is hope that our general approach to the problem can be adapted to this case. In this work, we took a physical model of the device and then derived a statistical model from it which we then fit to measurements. This approach allowed the statistical model to be interpretable and robust, because the learned parameters are connected to the original physical model, but it also allowed it to abstract away from most details of the capacitance matrices of the device. For curvilinear devices the first step would be to devise a sufficient generalization of the constant interaction model that still allows for efficient analysis to then derive a sufficient generalization of the statistical model. Of course, devising a proper model does not entail that we can learn or optimize it. For example, in curvilinear devices, the question of whether a set of points represents a strongly curved transition or two transitions with an intersection might become difficult to answer given noisy measurements.

Ultimately, we believe that not all devices can be tuned efficiently, due to the mathematical difficulty of the tuning task in its most general form. Our work puts very strong assumptions on devices and shows that when these assumptions are fulfilled, it is possible to find the transitions of interest, even for the largest devices manufactured to date. Future work then has the task to weaken the assumptions and show that tuning is still possible. We believe that weakening the assumptions on the regularity of the convex polytopes (e.g., as the ones obtained from simulated devices like [19]) are still within reach. Beyond that, we expect that the weakening

of assumptions comes at so large computational costs, that only smaller devices can be tuned, at which point layout and manufacturing strategies become more constrained towards devices that are still tuneable. We see our work as a first step along this line.

## Acknowledgements

This project has received funding from the European Union's Horizon 2020 research and innovation program under the Marie Sklodowksa-Curie grant agreement No. 895439 'ConQuER'.

## Code and Data Repository

The code (and link to the associated data) can be found at https://github.com/Ulfgard/quantum_polytopes.

## A  Sampling

There is a large variation of the size and surface area of facets in a Coulomb diamond. Thus, it is unlikely that we obtain enough samples for each facet using random sampling alone. Instead, for general $P_n$, we will obtain new measurements by constructing points for each candidate facet of our learned estimate $\tilde{P}_n$, through which we perform a line-search. To do this, we have to handle first that we only have learned a probabilistic model for $\tilde{P}_n$. Not all possible candidates of transitions $t_k$ have enough evidence in the model, which is represented by a small $\|W_k\|$. These facets might not exist, or there is not enough evidence in the dataset to support them. In both cases, the model will likely move the facets outside the polytope. Our approach is to take these facets, treat them as existing and move them back inside the polytope. This will likely generate an additional facet to sample new candidate points from. If the facet does not exist, these points will result in the facet being pushed further away, until eventually it can only be placed in a corner of the model, which rules it out.

This can be formalized as follows: If $\|W_k\| \geq 0.1/\delta$, we assume that there is sufficient evidence for it in the model and add the linear equation $W_k^T v + b_k \leq 0$ to $\tilde{P}_n$. If $\|W_k\| < 0.1/\delta$, we add a replacement facet $\tilde{W}_k^T v + \tilde{b}_k \leq 0$ with $\tilde{W}_k = W_k$ and $\tilde{b}_k = \max_i^\ell \tilde{W}_k^T v_i^-$. This moves the facet as much inside the polytope as possible without miss-classifying a point that is known to be inside $v^{-i}$.

Afterwards, we can sample points on each facet in $\tilde{P}_n$ by handling two cases:

- The facet belonging to transition $t_k$ intersects with the polytope $\tilde{P}_n$ in more than one point. In this case, we can compute the largest inscribed hypersphere on the facet (see Appendix C) in $\tilde{P}_n$ as a lower bound on the surface area covered by it. We then sample three points within the hypersphere uniformly at random. Sampling multiple points allows to quickly find enough points supporting a small facet in order to fulfill our stopping criterion, see Apendix B. If a facet is already supported by a large number of point pairs, we skip sampling from it in order to save measurement time.

- The facet belonging to $t_k$ does intersect with $\tilde{P}_n$ in at most a single point.

  In this case, we can not sample from the inscribed hypersphere, as the facet does not exist. Instead, we will find the closest point $v \in \tilde{P}_n$ to the facet and select it as candidate.

This point is the solution of the LP

$$\max_x W_k^T v$$
$$\text{s.t. } v \in \tilde{P}_n$$
$$\wedge\, v_i \geq l_i, i = 1, \ldots, G\,,$$

where $W_k$ is the normal of the facet belonging to $t_k$ in $\tilde{P}_n$ and $l_i$ the lower-bound in the $i$th gate voltage.

For each of these candidate points we conduct a line-search starting from an estimated mid-point of the polytope through the sampled points on the boundary. Each line-search returns another pair $(v^-, v^+)$ that we add to the dataset. To prevent that multiple copies of similar points are added to the dataset, we will add new points only if there are no points in the dataset within a distance of $\delta/4$.

For $P_0$ we decided to use a strategy that ensures that samples are spaced over a wide surface area of the polytope. Since in our experiments $G = N$, all facets intersect in a single point. Thus we can simply sample rays starting from that point along a facet until it hits the lower bound in any coordinate. Along this ray we can then sample a point on the boundary. Afterwards we conduct a line-search starting from the lower bound of voltages $-2\mathbb{1}_N$ to the selected point on the boundary. However, due to our use of a large number of initial points, this sampling strategy needed to be employed only rarely.

## B Termination condition for the algorithm

The stopping criterion of the algorithm is based on a check that for all facets the algorithm found, we either established that the facet is correct, or that it is too small to be estimated reliably with the line-search precision available.

We base the check for correctness of a facet on the fact that a plane in $G$ dimensions can be uniquely defined via $G$ linearly independent points it passes through. Our line-search procedure however, does not produce single points, but point pairs $(v_i^-, v_i^+)$, $i = 1, \ldots, \ell$ bounding the transitions of the polytope. While a single plane can pass through multiple points between a point-pair, finding more than $G$ point-pairs that are separated by the plane gives strong evidence that the facet found by the algorithm is real and that its parameters are correct.

If the facet is small in some direction, the limited precision of the line-search might make it impossible to reliably estimate its parameters or even disprove its existence. Thus, for each facet, we compute the radius $r$ of the largest inscribed hypersphere (see Appendix C). We only consider correctness of equations belonging to facets with radius $r > r_{\min}$ and consider facets smaller than that as undecided: the algorithm returns them, but does not claim that they are correct. In our work and evaluation, we consider these facets as non-existing/not found.

In our implementation, we chose $r_{\min} = 2\delta$. Then, for facets with $r > r_{\min}$, we compute the number of point pairs in the dataset separated by them. A facet with parameters $w, b$ separates a point pair $(v^-, v^+)$ if it holds $w^T v^- + b < 0$ and $w^T v^+ b > 0$. We consider a facet correct if more than $G + 3$ point pairs fulfill this condition.

To summarize, the stopping criterion of our algorithm is that for each facet of the polytope $P$, either $r < r_{\min}$ or it separates more than $G + 3$ point pairs.

## C  Computing the largest inscribed hypersphere

Given a $G$ dimensional polytope $P$ with linear inequalities $W_k^T x + b_k \leq 0$, $k = 1, \ldots, M$, the largest inscribed hypersphere $\mathcal{B}(m, r) = \{x \mid \|x - m\| \leq r\}$ with radius $r$ and midpoint $m$ is the solution of the problem

$$
\begin{aligned}
&\max_{r, m} \; r \\
&\text{s.t. } x \in P, \; \forall x \in \mathcal{B}(m, r) \\
&\qquad \wedge \, r > 0 \,.
\end{aligned}
$$

It can be computed as a solution to the equivalent LP

$$
\begin{aligned}
&\max_{r, m} \; r \\
&\text{s.t. } W_k^T m + b_k + \|W_k\| r \leq 0, \; k = 1, \ldots, M \,.
\end{aligned}
$$

In our application, we need to compute the largest inscribed hypersphere on the $i$th facet of $P$, which is a $G - 1$ dimensional object. To do this, we first compute the polytope of the facet $f_i = \{x \in P \mid W_i^T x + b_i = 0\}$ and find a $G - 1$ dimensional coordinate representation for $f_i$, before we can compute the largest inscribed hypersphere. For this, we first compute a rotation matrix $Q$, so that $W_i^T Q = (0, \ldots, 0, \|W_i\|)$, which can be achieved by defining $Q$ as a householder reflection. With this, we can substitute coordinates $x = Qz$, and obtain as the equality constraint of the $i$th facet $W_i^T x + b_i = W_i^T Q z + b_i = \|W_i\| z_G + b_i = 0$. Thus, in this coordinate system, we obtain immediately that $z_G = -b_i / \|W_i\|$ . This allows us to obtain the $G - 1$ dimensional description, by rewriting $f_i$ in terms of the coordinates in $z$ and removing the coordinate $z_G$

$$
\tilde{f}_i = \left\{ \tilde{z} \in \mathbb{R}^{G-1} \,\middle|\, \sum_{j=1}^{G-1} (W_k Q)_j \tilde{z}_j + b_k - (W_k Q)_G \frac{b_i}{\|W_i\|} \leq 0, k \neq i \right\} \,.
$$

With this representation, it is possible to compute the solution of the $G - 1$ dimensional inscribed hypersphere problem. And for any point $\tilde{z}$ in the inscribed sphere the corresponding $G$ dimensional coordinate becomes

$$
x = Q \left( \tilde{z}, -\frac{b_i}{\|W_i\|} \right)^T \,.
$$

## D  Derivation of the model

To derive our model, we start with a suitable multi-class classification problem, assuming first that in each line-search we can observe which transition $t \in T$ occurred. With this, we can assign each $v^+$ a label $y = 1, \ldots, N$ according to the index of $t$ in $T$. We will further assign the label $y = 0$ to all points $v_i^- \in P_n$, leading to $N + 1$ classes in total. We can now create a linear probabilistic classifier assigning class $y$ to point $v$ via

$$
p(y|v, W, b) = \frac{1}{Z(v, w, b)} \cdot \begin{cases} 1 & \text{if } y = 0 \,, \\ \exp(W_y^T v + b_c) & \text{if } y > 0 \,. \end{cases}
$$

Here, $W \in \mathbb{R}^{N \times G}$ is the vector of scaled transition normals and $Z(v, W, b) = 1 + \sum_{k=1}^{N} \exp(W_k^T v + b_k)$ is the normalization constant. This model is equivalent to multi-class

logistic regression and knowing the labels, the transitions can be learned easily via maximum-likelihood estimation.

In reality, we can not observe the label of the transition and instead only observe whether $y = 0$ or $y > 0$ by detecting the presence of a transition between a pair of points. This leads to a binary classification problem with probabilities $p(y = 0|v, W, b)$ and $p(y > 0|v, W, b) = \sum_{y=1}^{N} p(y|v, W, b)$. The model $h$ can be derived from this via

$$h(v) = \log \frac{p(y > 0|v, W, b)}{p(y = 0|v, W, b)} = \log \sum_{k=1}^{N} \exp(W_k^T v + b_k).$$

This model can be learned using maximum likelihood estimation. Assuming a set of point pairs $(v_i^-, v_i^+)$, $i = 1, \ldots, \ell$, we can assign to each point $v_i^-$ the label $y = 0$, as those points are inside the polytope by assumption. For the points outside, we can only assign that $y > 0$. With the previously defined probabilities, we obtain the regularized maximum likelihood objective

$$\max_{W,b} -\Omega(W, b) + \sum_{i=1}^{\ell} \log p(y = 0|v_i^-, W, b) + \log p(y > 0|v_i^+, W, b). \tag{D.1}$$

Here, $\Omega$ is a regularization term. For the training details of the final models trained that include the assumptions of the constant interaction model, see Appendix E.

# E  Regularization and Learning

## E.1  Learning $\Gamma$

For learning $\Gamma$, we substitute $W = \Gamma$ in equation (D.1) and pick a regularizer $\Omega$ which steers the optimizer away from some of the bad local optima using the prior knowledge of the task. We make use of two properties: a) for all transitions, we expect the absolute voltage value required to add an electron on the $k$th dot to be small and b) we expect that the searched normals are similar to the standard coordinate axis. Based on this, we propose the following regularizer:

$$\Omega(\Gamma, b) = \alpha_1 \|\Gamma^{-1} b\|^2 + \alpha_2 \sum_{k=1}^{N} \left(1 - \frac{\Gamma_{kk}}{\|\Gamma_k\|}\right)^2.$$

The first term computes the intersection of all boundaries and penalizes it based on the squared distance from the origin. The second term computes the cosine of the angle between the $k$th row of $\Gamma$ and the $k$th standard basis vector and penalizes the squared distance to $\cos(0) = 1$.

As parameters, we choose $\alpha_1 = 100$ (assuming that $\|\Gamma^{-1} b\|$ has units in Volt) and $\alpha_2 = 100$. We initialize the optimizer by choosing initial values $\Gamma = \frac{1}{\delta} I_N$ and set $b_k = -\frac{1}{\delta} \max_i^{\ell} W_k^T v_i^-$.

In the solver, we reparameterized $b = -\Gamma q$, which, when $\Gamma$ is invertible is equivalent to $q = -\Gamma^{-1} b$ as in the constraint. This makes solving slightly more numerically stable, as no inverse of $\Gamma$ needs to be computed in the regularizer.

## E.2  Learning $P_n$

For learning $P_n$, we substitute $W = c T \Lambda \Gamma$ in equation (D.1) due to our model assumption (4). As we keep $\Gamma$ and $T$ fixed, the regularized maximum-likelihood objective becomes

$$\max_{c,\Lambda,b} -\Omega(c, \Lambda) + \sum_{i=1}^{\ell} \log p(y = 0|v_i^-, c T \Lambda \Gamma, b) + \log p(y > 0|v_i^+, c T \Lambda \Gamma, b), \tag{E.1}$$

under the constraints that $c_{ii}, \Lambda_{ii} > 0$ and $\Omega(c, \Lambda)$ is again a regularisation term, as defined below.

We have a bit of knowledge about the parameters from our model that can help guide the optimizer to a reasonable solution. Since the devices we consider have a regular shape, the norms of $W_k$ should be similar. Therefore, $\lambda_{ii} \cong 1$. Further, if the $k$th facet exist, then $\|v_i^+ - v_i^-\| \leq \delta$ implies that $\|W_k\| = \mathcal{O}(\frac{1}{\delta})$. We can put both together using the regularizer

$$\Omega(c, \Lambda) = \sum_{i=1}^{N} \alpha_1 \delta^2 \|c_{ii} t_i^T \Lambda \Gamma\|^2 + \alpha_2 \log^2(\Lambda_{ii}).$$

The first term is a regularizer based on the squared norm of $W_k$ scaled by $\delta^2$ to reflect the expected scale. For candidate transitions that are not found by the optimizer, this term also drives the norm $\|W_k\|$ to zero, which gives an easy condition to filter out transitions which are not used by the model. The second term penalizes deviations of $\Lambda_{kk}$ from one on log-scale. This also prevents that the optimal solution of $\Lambda_{kk}$ can become close to zero or negative. Both terms are weighted by regularization factors $\alpha_1$ and $\alpha_2$. We found that $\alpha_1 = \frac{1}{100}$ and $\alpha_2 = 10$ worked well in practice.

Finally, in order to remove the positivity constraints from the problem, we performed a reparameterization: $c_{kk} = \log(1 + \exp(c_{kk}'))$ and $\Lambda_{kk} = \exp(\Lambda_{kk}')$ and then solved for $c_{kk}'$ and $\Lambda_{kk}'$ with the fitting reparameterized starting values.

## F  Device generation

In our experiments, we will simulate 3x2, 3x3 and 4x4 devices. For all devices, we choose the number of gates equal to number of dots, i.e., $G = N$. We further assume that the dominating capacitance of a dot is that of its plunger gate to the gate electrode. The parameter matrices $C^{DG}$ and $C^{DD}$ of an array of size $U \times V$ are randomly generated as follows. We create a connection matrix $C_0$ for the chosen device layout and create randomized $C^{DD}$ and $C^{DG}$ from $C_0$ via:

$$C^{DG} = S_g \left( I_N + \frac{\rho}{10} C_0 \right) + \epsilon,$$
$$C^{DD} = -\frac{\rho}{10} S^D C_0 S^D + Q.$$

Here, $I_N$ is the $N$ dimensional identity matrix, $S_g$ and $S^D$ are diagonal matrices with entries $\exp(z), z \sim \mathcal{N}(0, 0.01)$ and $\epsilon \in \mathbb{R}^{N \times N}$ is a noise matrix with entries $\epsilon_{ij} \sim \text{Uniform}(0, 0.02)$. The factor $\rho$ governs the interaction strength between dots and gates and finally, $Q$ is a diagonal matrix that is chosen such, that $\sum_j^N (C^{DD})_{ij} - \sum_k^G (C^{DG})_{ik} = 0$, $i = 1, \ldots, N$. The random variables $S_g$, $S^D$ and $\epsilon$ model deviations introduced by device manufacturing.

We compute $C_0$ as follows: We pick $(C_0)_{ij} = 1$ if array locations $i$ and $j$ are direct horizontal or vertical neighbours in the array grid. If they are direct diagonal neighbours, we pick $(C_0)_{ij} = 0.3$, and 0 otherwise. In our experiments, we pick $\rho \in \{1, 3\}$. The higher $\rho$ is, the less $C^{DD}$ resembles a diagonal matrix and the more the polytope becomes distorted.

## G  Computing the baseline

Computing the ground truth polytopes of a spin-qubit array based on the constant interaction model is a hard problem in higher dimensions. When computing the boundaries of a $N$-dimensional polytope $P_n$ formed by the constant interaction model where $D = N$, there are up

to $3^N - 1$ candidates for linear halfspaces $a_k^T v + b_k \leq 0$, $k = 1, \ldots, 3^N - 1$. This initial number can be reduced considerably in certain cases. For example, if a target state $n$ contains a dot with $n_i = 0$, we can ignore transitions that remove an electron at that location. Similarly, if the device is not connected to an external reservoir, we only need to consider transitions that leave the number of electrons in the array constant. From the remaining candidates, most will still not intersect with $P_n$ and must be filtered out.

To determine whether a candidate halfspace $a_l^T v + b_l \leq 0$ forms part of the boundary of $P_n$, we need to find a point $v \in P_n$ such that $a_l^T v + b_l = 0$. Since a point fulfilling $v \in P_n$ fulfills $a_k^T v + b_k \leq 0$, for all candidate halfspaces, determining the boundaries that make up a convex polytope can be done by solving a linear program (LP) for each candidate, where each LP has up to $3^N - 2$ linear inequalities. This naive approach is not feasible for the large devices considered in this work.

To reduce computation time, we will use a multi-stage approach for computing candidates. We use the fact that we can exclude the $l$th linear boundary, when we find any subset of candidate inequalities $k_1 \ldots, k_K$ such, that there exists no point $v \in \mathbb{R}^N$ that fulfills $a_l^T v + b_l = 0$ and $a_{k_i}^T v + b_{k_i} \leq 0$, $i = 1, \ldots, K$. Thus, if we find a set of likely candidates, we can quickly remove a large part of halfspaces via this approach by solving relatively small LPs. A suitable set of candidates are all transitions that add/remove an electron from the device, as well as all transitions where a single electron moves between dots.

We therefore partition the candidates into batches of 1000 inequalities, adding to each batch the likely transition candidates described above. We then solve an LP for each candidate in the batch. Afterwards, we iteratively merge all remaining candidates of all batches and solve the resulting LPs after each merge. We use this strategy to compute exact polytopes for $N \leq 9$.

For larger arrays, we compute polytopes by only computing transitions involving electron changes within sub arrays of shape $u_s \times v_s$, $u_s v_s \leq 9$. We consider all possible sub arrays of this shape that are contained with the device layout. For each sub array, we compute all candidate transitions and merge them, potentially adding additional candidates of interest. Afterwards, we use the algorithm above to compute the polytope. For example, in a 4x4 grid with $N = 16$, we create 3x3 sub arrays. There are 4 ways to fit a 3x3 array into the 4x4 grid, leading to $4 \cdot (3^9 - 1)$ candidate transitions. To this set we add all single electron transitions that are not contained in any of the blocks. While the number of candidates is still large, this strategy excludes a large number of transitions. For example, the computed ground truth will not contain any transition that involve simultaneous changes at more than 9 locations in the array.

# H  Runtime

Finally, we consider evaluation times and number of required line searches. For computing $\Gamma$, almost all trials finished in the first iteration of the algorithm, which means that computation time was in the order of seconds to minutes, while the number of line-searches was almost equal to the initial estimate. More interesting is the running time for the full target polytopes, which we depict in Figures 7a&7b.

Most noteworthy is the comparison of results in $S_4$ and $S_5$, which show the time saved by reducing the number of transiitons considered. The median number of line searches in these settings is reduced by 30-50%, while the running time is reduced by 50-70%.

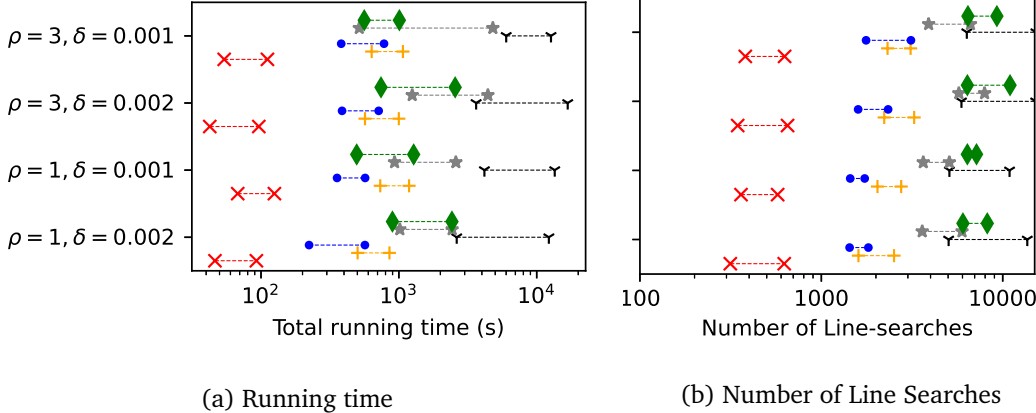

(a) Running time        (b) Number of Line Searches

Figure 7: Number of line searches and total running time for estimating the target polytopes in the different scenarios. Colors and symbols indicate the same scenarios as Figure 6.

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
