# Peer review of "Learning Coulomb Diamonds in Large Quantum Dot Arrays"

_SciPost Physics, doi:SciPost Phys. 13, 084 (2022)_

## Round 1 · Referee Report · Anonymous (Referee 1) · 2022-7-2

Strengths

1) The algorithm is elegant and performs well on benchmarks 2) Because the proposed approach is model-based it does not require training 3) The authors come up with clever heuristics to solve an otherwise prohibitively hard problem

Weaknesses

1) Some explanations could be improved 2) Lack of benchmarking with publicly available datasets 3) Extensibility to more realistic models will require significant work

Report

Operating semiconducting qubits requires first electrostatically defining quantum dots that confine a small number of electrons. While it is possible for a trained experimentalist to tune a small quantum dot array by hand, scaling to large processors will require automated procedures to perform state identification and tuning.

This work proposes a new state identification algorithm that learns the boundaries and transitions of a given charge state. Most of the available literature attempts to solve this problem with blackbox supervised ML algorithms. Here the authors take a model-based approach instead. In particular, they model a quantum dot array with a constant interaction (CI) model where each Coulomb diamond corresponds to a convex polytope. The task then becomes to then to learn the polytope equations with as few measurements as possible.

The algorithm works roughly as follows: first, the problem is recast as a binary classification task where the goal is to determine whether a given voltage point is in or out of the target polytope. Given a set of initial measurements, the algorithm uses regularized maximum-likelihood estimation to approximate the model parameters. Based on the fitted model, the algorithm proposes new measurements to refine the parameter estimation. The process is repeated until a desired precision/number of iterations is reached. While this estimation problem has been shown to be NP-hard, the authors come up with some clever heuristics that make it tractable (such as estimating the transition normals $W_k$ from the $n=0$ polytope). The authors benchmark the algorithm on 5 different scenarios and find it performs quite well.

Overall, I found the paper compelling and the benchmarks interesting. I congratulate the authors on a very nice work.

In terms of weaknesses, I would point out the following:

  1. Model-based approaches are appealing because they are interpretable and do not require training. However, they are manifestly less flexible than supervised ML approaches. As an example, the CI model fails to capture curvilinear transitions and excited levels inside a Coulomb diamond. I wonder: would it be possible to extend the model to account for curvilinear boundaries by e.g. making it quadratic in the voltages?

  2. The benchmarks are all performed on toy data. I think the paper would benefit from a comparison with publicly available datasets, such as doi:10.1371/journal.pone.0205844 and doi:10.5281/zenodo.2537934.

  3. Some explanations were a bit hard to follow, namely the discussion on page 6 about learning $\Gamma$ and $\Lambda$. Could the authors perhaps clarify it?

I would be happy to recommend publication after the authors address the concerns I outline below.

Requested changes

  1. As I mentioned previously, I think the explanation of the role of $\Gamma$ and $\Lambda$ could be improved.

  2. In my view, the paper would benefit greatly from including more visuals. For example, it would be helpful to see an algorithm run: the estimated model vs iteration, chosen line searches, etc. I realize this is nontrivial due to the high-dimensionality of the voltage space, but one could run the algorithm in some 2D voltage subspace.

  3. It seems to me that $W_k$ is used in the paper in two distinct ways. It is first introduced on page 3 to define the polytope equations. As far as I understand $W_k$ can be scaled arbitrarily (provided that $b_k$ is scaled accordingly). Later on page 5 $W_k$ is used as a parameter for logistic regression. There its norm matters because it determines the sharpness of the classifier (illustrated in Fig 3). Am I understanding this correctly? If so, perhaps the authors could clarify the notation and/or clarify this distinction.

  • validity: top
  • significance: high
  • originality: high
  • clarity: good
  • formatting: perfect
  • grammar: perfect

Author:  Oswin Krause  on 2022-08-01  [id 2700]

(in reply to Report 1 on 2022-07-02)

The review by referee #1 has been very helpful in improving the paper, and in pointing out omissions that we agree make the paper more accessible. We have incorporated the requested changes, in particular:

  1. We added a new paragraph explaining the role of \Gamma and \Lambda. Here is the text we added:

    “This decomposition is not strictly necessary from a mathematical point of view, since every change of the norm can be folded into Gamma by multiplying its rows with a diagonal entry of Lambda. However, during optimization, it allows us to differentiate learning the direction of the transitions (Gamma) and their norm (Lambda). In our approach, both aspects are handled independently, as we have different prior knowledge regarding the distribution of directions and their relative norm differences. We will describe this in more detail in the following.”

  2. New figures that highlight this have now been added.

    When creating the figures, we realized that we did not properly describe the part of the algorithm that connects the learned probabilistic model with the polytope: especially the handling of transitions with low evidence (small norm of W_k) during the sampling process. We amended the script in the appendix and main text. We further added a new section III.B where we discuss an example of a simulated DQD.

  3. We now clarify that while initially the norm of W_k was not of interest for the definition of the polytopes, now the model will (learn to) pick a choice of W with a large enough norm such that the resulting model is sharp.

In addition, regarding the question about curvilinear features (also asked in report #2), we have added two paragraphs in the discussion that link back to previous work in which this point is discussed in more detail.

We made it clearer that our algorithm is currently only theoretical: the assumptions we have on the device are very strong which makes it difficult to apply it to real dataset. This is also not our goal, because we want to see whether it is possible at all to tune large idealized devices. Our long term goal is to apply this to real devices, but we relay this to future work. We wrote a paragraph regarding this in the introduction and as the last paragraph in the discussion.

Please note that due to a comment of reviewer 2, we added a new scenario. As a result, we renamed S5 to S6 and added a new scenario S5 that investigates the reduction of line searches and running time when reducing the number of transitions T.

---

## Round 1 · Referee Report · Anonymous (Referee 2) · 2022-7-5

Strengths

In their paper “Learning Coulomb Diamonds in Large Quantum Dot Arrays”, Krause et al. have developed a novel method to initialise an array of qubits into a given charge state, which becomes heuristically impractical as devices scale up to the NISQ regime. The main novelty in their approach is limiting their search in voltage space to determining all the transitions of a specified charge configuration. This is achieved by using maximum likelihood estimation to determine the most probable convex polytope that describes the system.

The ability to automatically tune an array of quantum dots is a crucial challenge within the community, and most automated tuning techniques have currently been developed for a DQD system. As a result, I consider the results in principle to be suitable for publication in SciPost.

Weaknesses

The major limitation in determining the viability of this work is that it has not been tested on experimental data, not even for a DQD array. Therefore, it is unclear how sensitive the method is concerning potential experimental noise, such as charge noise and 1/f noise.
Another practical difficulty can arise from miss-calibrated PID control in a charge sensing set-up, which could also result in miss-labelled transitions. Hence, it is hard to gauge the reliability of the proposed protocol and the quality of the experimental data required. As such, I urge the authors to at least test their method on a DQD device for multiple charge configurations to determine a potential success rate.

Report

Another potential concern is the number of measurements required (15,000) to characterise a polytope. Could the authors explain how they arrived at such a figure and how the number of measurements would scale as a function of QDs? How long would it take to acquire and computationally process?

It would be beneficial for the reader if the introduction included an overview of how their algorithm would integrate into a fully automated tuning protocol. What conditions would have to be fulfilled before their algorithm gets implemented?

How do you get to the worst-case scenario of 3^N−1 facets? For a single QD should there not be four facets to create a Coulomb diamond, and six in a DQD to form the typical honeycomb pattern?

If only certain QDs in an array have access to a reservoir, how would one implement this algorithm to shuttle electrons across to an isolated pair of QDs? For instance, it is not clear to me how one would populate an isolated pair of QDs to the (3,1) -> (4,0) charge configuration?

For experiment S1, the authors should make it clearer that the electron pair have the same spin; otherwise, if the electrons had opposite spins, they could both populate the same QD.

The authors assume that charge transitions can be described by the constant interaction model, which is not necessarily valid, especially in the few-electron regime. How would the algorithm cope if the facets of the polytope were curved due to a large tunnel coupling?

In the evaluation of Gamma section, the authors claim that they obtained at least 17 successful trials for rho=3. Out of how many attempts?
Similarly, it is not clear how many trials were run for the evaluation of the P_n section.

Finally, there is a small typo present in the manuscript:

“This was achieved in all but 2 trials in the the setting rho = 1 and delta = 0.002”
  • validity: good
  • significance: high
  • originality: high
  • clarity: ok
  • formatting: reasonable
  • grammar: good

Author:  Oswin Krause  on 2022-08-01  [id 2699]

(in reply to Report 2 on 2022-07-05)

Referee #2 raises multiple concerning points, for which we are grateful. They have made an impact on the rigor of our manuscript. Our responses to the questions and concerns are listed below, in the order in which the referee raised them.

A major concern of the referee is that we have not managed to apply this algorithm to a real device. Ultimately, we believe that a variation of this algorithm will be able to run directly on a device, as was demonstrated in previous work (citation 9 in the manuscript). The algorithm may need a few modifications (additional terms in the definition of W_k that add individual deviations as well as regularisation terms for these components), as will the model to include more realistic settings, which will be addressed in future work. To clarify this, we added the the following text to the introduction: “The algorithm presented in this work is theoretical. Our goal is to answer the question whether the tuning problem of identifying desired single-and multi-electron transitions is reliably possible for large-devices that follow the constant-interaction exactly. This is a different starting point than the work in [9] that aimed to develop a practical algorithm that works on small devices that do not require exact adherence to the constant interaction model, but ultimately does not scale to the devices considered in this work.”. We also added a new paragraph to the end of the discussion where we repeat the sentiment and where we describe better how we position our work.

In the following, we discuss the other questions and issues the referee raised:

  1. “Another potential concern is the number of measurements…” Both number of line-searches and running times (without measurement time) are given in Figure 7. Due to the different sizes of our experiments this also gives an indication of scaling behaviour. While 15000 measurements for a 16D device sound very large for practical application, the number is small compared to the size of the problem. Our computed ground truth polytopes in 16D have around 3700 facets of which up to 1200 have a size larger than our size cut-off (smallest dimension > 2.0*delta). Around 100 of those are in the set of transitions we are interested in in our experiment. We write in the paper that naive evaluation of the solution, i.e., checking that each parameter of the linear equation is correct, requires 16 line-searches. This is also roughly what our stopping criterion uses, and thus puts a lower bound of the algorithm at 1600, less than a factor 10 of the 15000 reported (by using properties of the constant interaction model one could reduce the 1600 significantly by directly testing the parameters, but in the larger context of devices that do not quite follow the constant interaction model, we can not expect anything lower than 1600, which was the reason why we used this stopping criterion).

    However, as we do not know the 100 existing facets beforehand, the true number must be larger than that, as we have to look at the larger set of potential transitions (which in our case is around ~300 elements), which requires not only finding the existing transitions, but also ruling out the 200 transitions not in the set and not confusing them with the 3600 (1100) wrong candidates. Given this task, the factor 10 does not sound extremely far off. One could save some measurements by reducing the set of transitions we search for. This is because our algorithm produces a measurement for all candidates each iteration, and thus if we halve the number, there is hope that this also halves the number of measurements. In our 4x4 experiment we search for all possible single electron transitions, which also includes the transition top left corner -> bottom right corner. In a real device already jumps further than "the diagonal neighbour" might not be observable and thus we could reduce the number of candidates, but it is unlikely to reduce it more than a factor 2-3 (each dot has between 3 and 8 neighbours). We added a new experiment, where we further restrict transitions to direct neighbours to investigate this. The results show that the number of line-searches in this scenario is now in the worst case slightly over 10000 (and the running time is reduced to <1h). Note that as a result of this, we renamed scenario S5 to S6 and added the new scenario as S5.

    Regarding the measurement time: This depends on the desired line-search precision and the line-search strategy. Higher precision requires more time in a strategy that uses 1D raster scans, but one can obtain also very high accuracy by using an initial quick line-search followed by a high precision search around the location of the transition. In the related work (references 8 and 9 in the paper), an analog ramp with hardware trigger for a transition was used, and the reported line-search ramp time was chosen as 1 second, this would bring the total tuning time to around 6h (4h line-search and 2h optimization time) in S4(old S5) and 4h in S5 (the new scenario).

    We added parts of this into the discussion at spots where it fits.

  2. “It would be beneficial to the reader…”

    We added the following text to the introduction: “The starting point for our tuning problem is a device that has undergone initial tuning: barrier gate voltages are chosen such that individual dots have been formed and sensor dots are tuned to compensate for cross-talk from the individual gate voltages. Moreover, the state-space of the device is explored such that initial states of interest are found (e.g., one electron on each dot).”

  3. “How do you get the worst-case scenario…”

    The upper bound is not tight and assumes that on each location at most one electron can be added or removed at a time. In a DQD you typically have 6 transitions, which is 2 less than the naive upper bound of 8, because this naive bound includes the (-1,-1) and (1,1) transitions which are unphysical.

    In higher dimensions, the fraction of a-priori unphysical transitions (more than one electron added simultaneously without an electron leaving and vice versa) is 2*(2^N-N -1): there are 2^N-1 choices of t that only add electrons to the device. Of those, N are physical as they add only a single electron. Since the same holds for only removing electrons, we get a factor 2.

    At N=16, the relative difference between the number of a-priori physical transitions and our bound of 3^N - 1 is only 0.3%. With more information about the device layout it is likely possible to constrain this number further, but this would be a difficult task on its own.

  4. “If only certain QDs in an array…”

    We are not quite sure what the final example in the question refers to. We interpret it as “two dots in a DQD are isolated from each other but each has access to a reservoir”. In this case, there does not exist a transition (3,1)->(4,0) in the device. Instead, for each state, there just exist two transitions that add/remove single electrons to/from their dots and the transition (3,1)->(4,0) is a degenerate corner of the polytope. In this case, the correct behavior of the algorithm is to learn a polytope that has 4 transitions: one for addition/removal of an electron on each dot. There does not exist a direct transition to the (4,0) state and thus it will not be learned. In this case, the correct approach is to take one of the transitions: either adding an electron to dot 1 or removing one from dot 2 and re-run the algorithm in the newly entered state to find the transition that reaches the (3,1) state (one could also take a path through the corner if one is sure).

    In the broader scope of dots not being connected to the reservoir (like S5) a similar problem can be encountered: assuming that the second dot is not connected to the reservoir, going directly from the (3,0) to the (3,1) state requires a transition that does not exist. In this case, we think that it is the task of the experimenter to define a proper set of transitions, similar to how we chose a path of transitions in scenario S1, for example by first adding an electron to the first dot and then shuttling one to the second dot.

  5. “For experiment S1…”

    We have made it clearer that we propose the route for cases where electrons are not known to have different spins.

  6. “The authors assume that charge transitions…” We are looking at weakly coupled devices, where one can hope that transitions are approximately linear, even in the small electron regime (see e.g., [9] where the polytope of the (1,1) and (1,1,1) states were learned). We would not propose to use it in the strongly coupled regime. We added a discussion of the curvilinear case as a response to reviewer 1 to the discussion which hopefully also answers your question.

  7. “In the Evaluation of Gamma section…”

    All scenarios were repeated 20 times, which involves computing P_0 and P_n one after another, where P_n uses the result of P_0 if that run was successful. For evaluation of gamma, we picked the results of P_0 computed for scenarios S1, S3 and S4. Since S4 and S5 are run on the same device type and to not cherry-pick results, we reported the worse of both scenarios as the number of successes. " all our results reported for S4 and S5 had at least 17 successful trials for ρ = 3". We added “(out of 20 trials)” at the end to clarify the number of trials.

---

## Round 2 · Referee Report · Anonymous (Referee 1) · 2022-8-2

Report

The authors replied to all my comments convincingly and I am happy to recommend publication.

---

## Round 2 · Author Response

We have tried to answer and in integrate all reviewer comments, we further added a number of changes we deemed necessary after integrating the changes, especially of some shortcomings we found ourselves.

---

## Round 2 · List of Changes

1. Added paragraph in the introduction clarifying at which point of tuning our algorithm becomes relevant
  2. Added paragraph in the introduction clarifying that the algorithm presented is theoretical and will not work on real devices
  3. Added new Figure 4, including a step-by-step visualisation of the algorithm on a DQD, answering requested change 2 of referee report 1. Further added section III.B that discusses the Figure as part of the example.
  4. AS a result of 3. we realized we did not describe a part of the algorithm in sufficient detail: added paragraphs in section III and Appendix A
  5. Clarified the differences between the two uses of W_k in section III.A, answering requested change 3 of referee report 1.
  6. Clarified the distinction of Lambda and Gamma in Section III.A
  7. As a result of answering a point of referee report 2, we added a new experiment, as variation of Scenario S4 where we reduces the number of transitions we search for to the most relevant ones (8 neighbourhood of any given dot). We renamed the old scenario S5 to S6 and added the new scenario as S5.
  8. Added various graphs to the discussion: Better embedding of prior work [8,9], used [8,9] to argue that the strong assumption of linearity is relevant in actual devices, discussion of curvilinear devices and outlined future work and work diections that better embed our work in the general domain.

---

## Editorial Decision

published